# Revealing the Therapeutic Potential of Muscle-Derived Mesenchymal Stem/Stromal Cells: An In Vitro Model for Equine Laminitis Based on Activated Neutrophils, Anoxia–Reoxygenation, and Myeloperoxidase

**DOI:** 10.3390/ani14182681

**Published:** 2024-09-14

**Authors:** Didier Serteyn, Nazaré Storms, Ange Mouithys-Mickalad, Charlotte Sandersen, Ariane Niesten, Julien Duysens, Hélène Graide, Justine Ceusters, Thierry Franck

**Affiliations:** 1Department of Equine Clinical Sciences, University of Liège, 4000 Liège, Belgium; nazare.storms@uliege.be (N.S.); charlotte.sandersen@uliege.be (C.S.); 2Center for Oxygen Research and Development, B6, University of Liège, FARAH, Quartier Vallée 2 Avenue de Cureghem 5D, 4000 Liège, Belgium; amouithys@uliege.be (A.M.-M.); ariane.niesten@uliege.be (A.N.); julien.duysens@student.uliege.be (J.D.); helene.graide@uliege.be (H.G.); j.ceusters@uliege.be (J.C.); t.franck@uliege.be (T.F.)

**Keywords:** horse, laminitis, metabolism, keratinocyte, mesenchymal stem cell, mitochondria

## Abstract

**Simple Summary:**

Equine laminitis is a serious condition causing severe pain and lameness in horses, often leading to euthanasia. This study developed a lab model to better understand laminitis using keratinocytes which were exposed to conditions simulating the disease. This research showed that adding muscle-derived stem cells helped protect and restore the keratinocytes’ metabolism. These results highlight the potential of stem cell therapy in treating laminitis, offering new hope for managing this debilitating disease in horses.

**Abstract:**

Laminitis in horses is a crippling condition marked by the deterioration of the dermal–epidermal interface, leading to intense lameness and discomfort, often necessitating euthanasia. This study aimed to establish an in vitro model of laminitis using a continuous keratinocyte cell line exposed to anoxia–reoxygenation and an activated neutrophil supernatant. A significant decrease in the keratinocytes’ metabolism was noted during the reoxygenation period, indicative of cellular stress. Adding muscle-derived mesenchymal stem/stromal cells during the reoxygenation demonstrated a protective effect, restoring the keratinocytes’ metabolic activity. Moreover, the incubation of the keratinocytes with either an activated neutrophil supernatant or myeloperoxidase alone induced increased keratinocyte myeloperoxidase activity, which was modulated by stem cells. These findings underscore the potential of muscle-derived mesenchymal stem/stromal cells in mitigating inflammation and restoring keratinocyte metabolism, offering insights for future cell therapy research in laminitis treatment.

## 1. Introduction

Laminitis, a prevalent condition in horses, is marked by the degradation of the dermal-epidermal junction, which can cause instability and the potential displacement of the distal phalanx. This pathological process frequently results in severe lameness and acute pain, occasionally leading to euthanasia [1,2,3]. Although the precise etiology remains incompletely understood, evidence suggests that laminitis arises from local manifestations of an exaggerated systemic inflammatory response, leading to a compromised blood flow, foot inflammation, endothelial/vascular dysfunction, extracellular matrix degradation, and metabolic perturbations in keratinocytes [4,5,6].

The early involvement of polymorphonuclear neutrophils (PMNs) in the onset of laminitis is well established in the literature [7,8]. In 2007, Riggs et al. demonstrated the presence of myeloperoxidase (MPO), a major protein from the alpha granules of neutrophils, in the bloodstream, skin, and lamellar tissue of horses with black walnut extract (BWE)-induced laminitis [9]. Furthermore, recent evidence shows the presence of MPO in lamellar tissue of clinical cases suffering from severe laminitis [10]. MPO is a heme-containing peroxidase that is expressed mainly in neutrophils and to a lesser degree in monocytes. In the presence of hydrogen peroxide and halides, MPO catalyzes the formation of reactive oxygen intermediates, including hypochlorous acid (HOCl). MPO has been demonstrated to be a local mediator of tissue damage and the resulting inflammation in various inflammatory diseases [11].

Even in cases where mitochondrial dysfunction has not been directly observed in the hoof, an energy deficiency has been proposed as a potential cause for the disruption of hemidesmosomes, which may contribute to the failure of the dermal–epidermal junction [12]. The relationships between inflammation, particularly neutrophil activation, either associated with ischemic reperfusion events or not, and mitochondrial dysfunction, are documented across various organs such as the heart, kidneys, and liver [13].

In 2014, a clinical study demonstrated a marked reduction in muscle mitochondrial oxidative phosphorylation in horses suffering from acute laminitis of various etiologies when compared to the levels observed in both fit and obese healthy horses [14]. Recently, He et al. [15] demonstrated that neutrophil extracellular traps (NETs) induced mitochondrial dysfunction in cardiomyocytes, associated with an increase of reactive oxygen species (ROS) release.

Given that neutrophil activation is observed in the early stages of laminitis and that we have recently identified the presence of neutrophil extracellular traps (NETs) in severe clinical cases, we hypothesize a potential correlation between neutrophil activation, MPO activity, NET release, and metabolism dysfunction in equine laminitis [10].

NETs are extracellular structures composed of decondensed DNA strands intertwined with histones and neutrophil granule proteins, such as MPO and elastase, which are released from neutrophils to trap and neutralize microbes. However, excessive NET formation during uncontrolled inflammatory responses has been implicated in the development of thrombosis and multiple organ failure in sepsis [16,17,18]. In an ischemic-reperfusion injury, Zhang et al. [19] reported that MPO-DNA complexes were the more cited markers of NET induction.

In sepsis, a recent metabolomic study confirmed that among patients with septic shock, non-survivors had a deeper and more persistent dysregulation of protein analytes, attributable to neutrophil activation and the disruption of mitochondrial metabolism, than the survivors [20].

Ischemic-reperfusion injury (IRI) takes place during reperfusion by activating inflammation and ROS production, causing mitochondrial damage and apoptosis of parenchymal cells [21]. Mesenchymal stem/stromal cells (MSCs) for the treatment of multi-organ IRI is currently considered a valid approach to reducing the injury [22]. Moreover, extracellular vesicles (EVs) derived from human umbilical cord MSCs attenuate rat hepatic IRI by suppressing oxidative stress and the inflammatory response of neutrophil [23].

In 2017, Magana-Guerrero [24] demonstrated that amniotic-derived MSCs could interfere with the NET release by neutrophils via a mitochondrial pathway. In a previous study we described a minimally invasive technology to obtain muscle-derived MSCs (mdMSCs). These cells have similar properties to the MSCs from other sources and showed potent immunomodulatory effects [25].

Preliminary clinical studies using MSCs in laminitis show encouraging results. The therapeutic potential is attributed to the unique properties of the MSCs that target damaged tissues, inhibit the immune and inflammatory response, and facilitate repair [26]. However, the mechanisms of action could be deeply investigated.

Considering the aforementioned points, the objectives of this study were to generate an in vitro model of laminitis using keratinocytes that were exposed to anoxia–reoxygenation (A/R) in conjunction with an activated neutrophil supernatant and to reveal the potential therapeutic effect of mdMSCs on the keratinocytes’ metabolism and the MPO activity.

## 2. Materials and Methods

### 2.1. Chemicals and Reagents

The analytical grade phosphate salts, sodium and potassium chloride, sodium hydroxide, sodium acetate, H_2_O_2_ (30%), dimethyl sulfoxyde (DMSO), Percoll (GE Healthcare, Machelen, Belgium), hematoxylin solution modified according to Gill III (Merck), T-flasks, and conical-bottomed centrifuge tubes were purchased from Merck (VWR International, Leuven, Belgium). The bovine serum albumin fraction V (BSA) was obtained from Roche Diagnostics (Mannheim, Germany). The cytochalasin B (CB), N-formyl-methionyl-leucyl-phenylalanine (fMLP), sodium nitrite, cisplatin, and Triton X-100 were purchased from Sigma-Aldrich (Bornem, Belgium). The 96-well microtiter plates, Amplex red^®^ (10-acetyl-3,7-dihydroxyphenoxazine) (Invitrogen, Thermo Fischer (Waltham, MA, USA), paraformaldehyde solution 4% in PBS (Thermo Scientific, Waltham, MA, USA), trypsin TrypLE Express (Gibco, Thermo Fisher Scientific, Waltham, MA, USA), Hank’s balanced salt solution (HBSS) 1× (Gibco, Thermo Fisher Scientific, Waltham, MA, USA), and Fetal Bovine Serum (FBS) were purchased from Fischer Scientific (Merelbeke, Belgium). The Dulbecco’s Modified Eagle Medium Ham’s F12 (DMEM F12) culture medium with Hepes and glutamine, penicillin-streptomycin, amphotericin B, and the Dulbecco’s phosphate buffered saline (DPBS) were purchased from Lonza (Verviers, Belgium). The purified equine neutrophil MPO was characterized by a specific activity of 70.4 U/mg and a protein concentration of 3.38 mg/mL. The antibodies against equine MPO, derived from rabbits and guinea pigs, were sourced from Bioptis (Vielsalm, Belgium). Additionally, rabbit polyclonal antibodies targeting citrullinated Histone H3 (citrulline R2 + R8 + R17) were acquired from Abcam (Cambridge, UK).

### 2.2. Cells

The keratinocytes (HaCaTs) and muscle-derived mesenchymal stem cells (mdMSCs) were purchased from ATCC (USA) and Revatis SA (Aye, Belgium) and were cultured in DMEM high glucose with (10% FBS) and in DMEM-F-12 (20% FBS), respectively.

### 2.3. The Preparation and Characterization of the Activated Neutrophil Supernatant

#### 2.3.1. Equine neutrophils were obtained from blood collected by jugular venipuncture on 5 healthy horses using ethylene diamine tetraacetic acid (EDTA) tubes, as described by Pycock et al. [27]. Briefly, the neutrophils were isolated at room temperature (18–22 °C) by centrifugation (400× *g*, 30 min at 20 °C) on a discontinuous percoll density gradient. The polymorphonuclear fraction was collected in DPBS, counted, and diluted into DMEM high glucose + 10% FBS to obtain a suspension of 2 million neutrophils/mL. The neutrophils were stimulated by adding 1 µL of cytochalasin B (5 mg/mL) per ml of cell suspension in the medium and incubating them for 30 min at 37 °C. Thereafter, 10 µL of fMLP (10^−4^ M) per ml of cell suspension was added and cells were incubated for 30 min at 37 °C. The control conditions were performed in parallel with cell suspensions without the addition of CB and fMLP or with the replacement of the stimulating molecules by DMSO, the solvent used for their solubilisation (Ctrl DMSO). Finally, the cell suspensions were centrifuged for 5 min at 600× *g* and the supernatants were collected and stored at −20 °C for future experiments. The activated neutrophil and non-activated neutrophil supernatants were called ANS and NANS, respectively

#### 2.3.2. The active free MPO and the active MPO bound to the NETs were measured in the supernatants (NANS and ANS) after neutrophil incubation and stimulation, according to the techniques described by Storms et al., 2024 [10]

#### 2.3.3. The Measurement of Active MPO by SIEFED

The SIEFED (Specific Immuno-Extraction Followed by Enzymatic Detection) technique employs microplate wells coated with an immobilized primary antibody (polyclonal rabbit anti-MPO IgG). The undiluted supernatant was added to the wells and incubated for 2 h at 37 °C in the dark, facilitating the capture of MPO by the antibodies. Following sample removal and three washing steps, the substrate (H_2_O_2_) and co-substrates (nitrite and Amplex Red) were introduced to detect the peroxidase activity of MPO, which was indicated by the oxidation of Amplex Red to its fluorescent product, resorufin. In more detail, the peroxidase activity of MPO was monitored by adding 100 µL of a 40 µM Amplex red solution that was freshly prepared in a 50 mM phosphate buffer, pH 7.4, and supplemented with 10 µM H_2_O_2_ and 10 mM sodium nitrite. The fluorescence was measured at excitation and emission wavelengths of 544 nm and 590 nm, respectively, over 30 min at 37 °C using a fluorescent plate reader (Fluoroskan Ascent, Fisher, Merelbeke, Belgium). The fluorescence intensity was directly proportional to the amount of active MPO present in the sample. The MPO concentrations were determined by referencing a calibration curve generated with purified equine MPO, ranging from 2 to 140 ng/mL.

#### 2.3.4. The NET-Bound-MPO Activity

The MPO was measured in the supernatant after neutrophil incubation and stimulation, according to the techniques described in Storms et al. [10]. Neutrophil extracellular traps (NETs) were captured using anti-histone H3 (citrulline R2 + R8 + R17; anti-H3Cit) antibodies. The presence of active MPO bound to the NETs was subsequently detected using the same method as in the SIEFED assay. This involved coating a transparent 96-well microplate with immobilized primary rabbit anti-H3Cit antibodies (0.5 μg/mL) diluted in 20 mM phosphate-buffered saline (PBS). Following the removal of the coating solution, the plates were incubated for 150 min at 22 °C with a blocking buffer (PBS with 5 g/L of BSA) and then washed four times with a PBS containing 0.1% Tween 20. The plates were dried for 3 h at 22 °C and stored in a dry environment at 4 °C until use. The samples were loaded into the anti-H3Cit-coated wells in duplicate and incubated for 2 h at 37 °C. After the supernatants were removed and the wells were washed four times with PBS containing 0.1% Tween 20, the active MPO was measured. To reveal MPO peroxidase activity bound to the NETs, sodium nitrite and Amplex Red were added as described in the SIEFED assay, and the fluorescence was measured over 30 min using a fluorescent plate reader (Fluoroskan Ascent, Fisher, Merelbeke, Belgium). The level of active MPO bound to the NETs was evaluated using a calibration curve, ranging from 2 to 140 ng/mL, created with purified equine MPO in wells coated with polyclonal rabbit anti-MPO IgG antibodies.

### 2.4. The Effect of the NANS and the ANS on Hacat Metabolism in Normoxia and in Anoxia

HaCaTs were seeded in 2 transparent 96-microwell plates (10,000 cells/well) for 24 h in DMEM high glucose with (10% FBS) to let them adhere. Afterwards, the medium was removed, the wells were washed once with HBSS, then the plate was divided into 5 parts to add either the medium alone, the medium containing cisplatin (1.10^−3^ M) as cytotoxic control, or the mediums prepared with the non-activated (NANS) or the activated (ANS) neutrophil supernatants. A control assay was performed with the supernatant obtained with neutrophils in the presence of DMSO (Ctrl DMSO). In all the tested conditions, the HaCaT medium was used without red phenol. One plate was exposed to anoxia with a controlled oxygen level at 0.2% and the other one to normoxia (18.6% oxygen) for 48 h. After incubation, the plate incubated under anoxia was transferred to normoxia with the other plate and incubated for 24 h. The medium from the wells of the two plates was removed and the wells were washed once with HBSS before the addition of 100 µL HBSS and 10 µL MTS (3-(4,5-dimethylthiazol-2-yl)-5-(3-carboxymethoxyphenyl)-2-(4-sulfophenyl)-2H-tetrazolium). The MTS used as metabolic viability-based assays was from Promega (CellTiter 96 Aqueous non-radioactive cell proliferation assay^®^). Just after the MTS’s addition, a first reading of the absorbance at 490 nm was made with a Multiskan Ascent spectrophotometer (Labsystem, Thermo Fischer, Waltham, MA, USA); then, this was repeated every 1 h until 4 h to follow the evolution of the absorbance.

### 2.5. The Effect of mdMSCs on HaCaT Metabolism

HaCaTs were seeded as above in two transparent plates for 24 h, but some wells were left empty. At cell adherence, the medium was removed, the cells were washed once with HBSS, then the wells containing HaCaTs were divided into 3 parts to add either the medium (without red phenol), the medium containing cisplatin (1.10^−3^ M) as a cytotoxic control, or the medium prepared with activated neutrophils (ANS). One plate was exposed to anoxia with a controlled oxygen level at 0.2% and the other one to normoxia (18.6% oxygen level) for 48 h. Afterwards, the medium from the wells was removed and the wells were washed once with HBSS before the addition of 20,000 mdMSCs/well in all the wells containing HaCaTs to achieve a 50% HaCaT medium and 50% mdMSC medium ratio. In the wells previously left empty, mdMSCs were also added in the presence of the mixed medium to create a control with mdMSCs alone. The plates were then placed in normoxia for 24 h. Subsequently, the medium from the wells of the two plates was removed and the wells were washed once with HBSS before the addition of 100 µL HBSS and 10 µL MTS for the measurement of cell metabolism. Just after the MTS’s addition, a first reading of the absorbance at 490 nm was made with a Multiskan Ascent spectrophotometer (Thermo labsystem, Finland); then, this was repeated every 1 h until 4 h to follow the evolution of the absorbance. For the results’ interpretation, the metabolic response of the mdMSCs alone was subtracted from that of the HaCaTs + mdMSCs.

### 2.6. HaCaT–MPO Activity and Immunolocalization

#### 2.6.1. HaCaT Incubation with the ANS

The experiment was performed with a 6-well cell culture plate (CellStar, Greiner bio-one, Thermo Fischer, Whaltam, MA, USA) seeded with 100,000 HaCaT cells per well for 24 h in DMEM high glucose with (10% FBS) to let them adhere. After 24 h seeding, the HaCaT cells formed colonies that had a squamous appearance. The ANS obtained after CB/fMLP was diluted 4 or 8 times with the HaCaT medium and then added into the wells to give a final volume of 2 mL. Each dilution was tested twice. In the two remaining wells, none of the ANS was added. Following the addition of ANS, the HaCaT cells were incubated for 2 h at 37 °C with 5% CO_2_. After incubation, the medium was removed, and the wells were rinsed three times with 1.5 mL DPBS before measuring the in situ MPO activity.

#### 2.6.2. HaCaT Incubation with MPO

The experiment was performed with a 6-well cell culture plate (CellStar, Greiner bio-one) seeded with 100,000 HaCaT cells per well for 24 h. To the HaCaT adherent cells, a 100 µg/mL stock solution of equine MPO was added, resulting in two wells having 250 ng/mL MPO and two other wells having 500 ng/mL MPO. In the two remaining wells, no MPO was added. After the MPO’s addition, the HaCaTs were incubated for 2 h in the incubator (37 °C, 5% CO_2_). After incubation, the medium was removed, and the wells were rinsed three times with 1.5 mL DPBS before measuring the in situ MPO activity or performing the immunological detection of MPO.

#### 2.6.3. The Measurement of the HaCaT–MPO Activity

After washing, the in situ peroxidase activity of the MPO was monitored by adding 1 mL of a 40 μM Amplex red solution that was freshly prepared in 50 mM of a phosphate buffer, pH 7.4, supplemented with 10 μM H_2_O_2_, and 10 mM sodium nitrite. The fluorescence development was monitored during 30 min (37 °C) with a Fluoroskan Ascent (Thermo Fisher Scientific, Waltham, MA, USA) set at 544 nm and 590 nm for the excitation and emission wavelengths, respectively. The total fluorescence was directly proportional to the amount of active MPO in the sample.

#### 2.6.4. The Detection of HaCaT–MPO by Immunocytology

After incubating, the HaCaT cells with human MPO and/or mdMSCs were described, the medium was removed, the wells were washed three times with 1.5 mL DPBS, and the cells were fixed with cold 4% PFA solution (Thermo Scientific) for 10 min. Thereafter, a cell permeabilization was done by incubating the cell for 10 min with 0.1% Tritron X-100. After 3 washings with 1.5 mL DPBS, a circular zone at the bottom of the wells was delimited with a hydrophobic slide marker. This zone was immuno-stained for human MPO using the horseradish peroxidase/diaminobenzidine ABC detection immunohistochemistry (IHC) kit (Abcam), according to the protocol with slight modifications. A positive staining to the MPO was detected by a brown coloration. Successively, the steps were hydrogen peroxide block (10 min, RT), protein block (10 min, RT), incubation (1 h, 37 °C) with the rabbit primary antibody against human MPO (Abcam 9535, diluted 1:100 × with 20 mM PBS pH 7.4 + 0.5% bovine serum albumin and 0.1% Tween 20), biotinylated goat (15 min, RT), streptavidin peroxidase (15 min, RT), DAB chromogen substrate (10 min, RT), and, lastly, hematoxylin solution (45 s). Between each step, 3 washings with DPBS were performed (3 × 3 min). Two ml of DPBS buffer were added before the observation with a Carl Zeiss Axioskop 20 for transmitted light and the incident-light fluorescence microscopy connected to a digital camera (Nikon D70) were performed. All the photographs were obtained using the same light intensity and shutter speed.

### 2.7. The Effects of mdMSCs on HaCaT–MPO Activity

As in point 5.2, the experiment was performed with a 6-well cell culture plate (CellStar, Greiner bio-one) seeded with 100,000 HaCaTs per well. The adherent cells were preincubated with 250 ng/mL and 500 ng/mL equine MPO for 2 h, then the excess of MPO was removed by 3 washings with PBS. Following the washing step, a mix (*v*/*v*, 1/1) of DMEM high glucose and DMEM-F12 was added into the 3 upper wells, while in the 3 bottom ones, 200,000 mdMSCs in 1 mL DMEM-F12 were added to the HaCaT cells that were previously covered by 1 mL DMEM high glucose. Thereafter, the culture plate was incubated for 24 h in the incubator (37 °C, 5% CO_2_). After the removal of the medium, the wells were washed 3 times with 1.5 mL DPBS before the measurement of the in situ MPO activity.

### 2.8. Statistical Analyses

Six to fifteen independent experiments were conducted for the statistical analyses and the figure generation for the MTS assay and the MPO activity. Since the data did not conform to a Gaussian distribution, Kruskal–Wallis tests followed by post-hoc tests were employed to evaluate the impact of the different conditions (factors) on the MPO activity or MTS assay values (variable). The significance level was set at *p* < 0.01 (Medcalc software, Ghent, Belgium). The data in the figures are presented as medians with 95% confidence intervals and are expressed as relative values (%) compared to the control groups, which were standardized to 100% for the MTS assays.

## 3. Results

### 3.1. Free MPO and NET-Bound MPO Released by Neutrophils

The concentration of free active MPO and NET-bound MPO significantly increased in the ANS compared to the NANS. Additionally, active MPO appeared to be primarily associated with NETs (Figure 1)

### 3.2. The Effect of Normoxia and Anoxia on HaCaT Metabolism in the Presence of Neutrophil Supernatants

HaCaT cells exposed to anoxia-reoxygenation in the presence of the ANS exhibited a significant decrease in their metabolic activity compared to the control (normoxia) and the non-activated condition. Cisplatin used as a positive toxicity control showed a strong decrease in metabolic activity (Figure 2).

### 3.3. Effect of mdMSC on HaCaT Metabolism Submitted to Normoxia or Anoxia with the ANS

MdMSCs cocultured with HaCaTs showed a significant increase in the HaCaT metabolism in normoxia and a protective effect against the stress conditions induced by A/R in the presence of the ANS (Figure 3).

### 3.4. The Active MPO from the ANS and the Purified Equine MPO Are Captured by HaCaT

Despite the washing of the HaCaT cells exposed for 2 h with the ANS or the purified MPO, in situ peroxidase activity was measured. This activity appeared more dose-dependent for the purified MPO than for the ANS (Figure 4). The capture of MPO by HaCaTs was confirmed by IHC, as shown in Figure 5. After the pretreatment of the HaCaT cells by MPO and their immunolocalization by the anti-MPO antibody and the Abcam detection IHC kit, we observed an intense brown staining both in the cytosol and in the perinuclear zone of the cells (Figure 5D). The perinuclear staining suggested that part of the enzyme had entered the cells. A slight brown coloration was, however, also observed in the cells that were not treated with MPO, suggesting a slight unspecific staining (Figure 5C). In the absence of the anti-MPO antibody, no brown coloration was observed (Figure 5A,B).

### 3.5. MdMSCs Added to Pre-Treated MPO HaCaT Cells Inhibit the In Situ Activity of MPO

After the washing of the HaCaTs that were pre-treated for 2 h with MPO, mdMSCs were added to the HaCaTs and cultured for 24 h to let them adhere. After this incubation period, the mixture of the adherent cell populations (HaCaT + mdMSC) was washed three times with PBS and the in situ activity of the MPO was revealed as above. First, we noticed a strong decrease in the in situ MPO activity 24 h after the HaCaT’s initial contact with MPO. This decrease appeared to be higher than 80% for the HaCaT cells that were pre-treated with 500 ng/mL MPO. An additional decrease in MPO activity was observed when the HaCaTs were cocultured with mdMSCs, compared to HaCaT cells alone. This decrease appeared significant (*p* < 0.01) when the HaCaT cells were exposed to 500 ng/mL equine MPO (Table 1).

## 4. Discussion

The primary objective of this study was achieved. We successfully developed a straightforward in vitro model to explore laminitis by subjecting a continuous-keratinocyte cell line (HaCaT) to an ANS, following a period of anoxia and subsequent reoxygenation. The MTS tetrazolium assay was used in this study to evaluate the activity of cellular metabolism by measuring the activities of mitochondrial NAD(P)H oxidoreductases or cytoplasmic esterases [28,29]. This test is currently one of the most widely used methods to assess drug toxicity [30]. In our study, cisplatin was used as a cytotoxic molecule control to evaluate the stress intensity.

When cells were exposed to the ANS and anoxia followed by reoxygenation, a significant decrease in HaCaT metabolism was observed. Such a decrease was not observed in the cells that stayed in normoxic conditions. The ANS was prepared using a combination of CB and fMLP to simulate a more physiological process for neutrophil stimulation and MPO degranulation. Our results showed that the ANS contained an important concentration of free active MPO but also a more important NET-bound active MPO, in comparison to the NANS, which did not.

Herein, the inhibition of HaCaT metabolism upon exposure to the ANS was observed specifically following a period of anoxia followed by reoxygenation. It is widely recognized that IRI arises from mitochondrial dysfunction, leading to the production of ROS such as superoxide anion, hydroxyl radical, and hydrogen peroxide [31,32]. The latter serves as a substrate for MPO, generating additional potent oxidants that can contribute to the exacerbation of mitochondrial dysfunction. It is important to note that the ANS likely contained various other products released by activated neutrophils, such as cytokines, lipid mediators, and other proteolytic enzymes, that were able to participate in tissue injury [33]. In addition, He et al. [17] have indicated that NET release can also disrupt the mitochondrial function of targeted cells.

Interestingly, the addition of mdMSCs during the reoxygenation phase exhibited a protective effect, leading to a restoration of the cellular metabolic activity in the HaCaT cells treated with the ANS and previously submitted to anoxia, suggesting a beneficial effect of mdMSCs in the restoration of mitochondrial function during the stress condition, as often described in the literature [34]. However, an increase in mitochondrial function was also observed in the cells submitted to the ANS without an anoxia period. Indeed, mitochondrial transfer was described under both physiological and pathological conditions. MSCs can transfer entire mitochondria or fragments via extracellular vesicles (EVs) or nanotubules [35]. The mdMSC’s mechanism of action may involve the modulation of cellular respiratory function. Indeed, the preliminary results presented at the international conference, Targeting Mitochondria, which took place in Berlin in 2023 [36], suggest the ability of mdMSCs to transfer mitochondria in HaCat cells. However, the recovery of metabolic activity could also be due to the effect of mdMSCs on the NETs present in the medium. Magana-Guerrero et al. [27] demonstrated that amniotic-derived MSCs inhibited NET release by interfering with neutrophil mitochondria. Furthermore, their study showed that the inhibition of NETs, the reduction in ROS levels, and the loss of mitochondrial membrane potential were reversed by MSCs overexpressing TSG-6, an important factor that inhibits the TNF/NF-kB signalling pathway.

In addition to its pivotal role in the regulation of energetic metabolism, the administration of MSCs is increasingly recognized as a valid strategy for treating multi-organ IRI, the attenuating cytokine storms associated with acute respiratory distress syndrome (ARDS), systemic inflammatory response syndrome, and sepsis [22]. These beneficial effects are attributed in part to the immunomodulatory properties of mdMSCs. Our previous studies have demonstrated that in vitro, mdMSCs exhibit a significant decrease in neutrophil modulation involving the inhibition of ROS production, active MPO release, and NET-bound MPO activity [29].

Our finding suggested that active MPO played a significant role in the pathogenesis of laminitis. We observed its presence in the lamellar tissue of horse hooves that were submitted to experimental laminitis and in a clinical case where active NET-bound MPO was also measured and detected in lamellae [9,18]. Thus, using an in vitro cell culture model, we investigated whether MPO could enter into keratinocytes cells and if the supplementation of mdMSCs in this model could modulate the activity of MPO. We first demonstrated that the equine MPO as well as the MPO from the ANS were captured by the HaCaT cells and that this MPO remained active. IHC confirmed the presence of MPO intracellularly, mainly in a perinuclear manner.

Remarkably, the coculture of HaCaTs over 24 h with mdMSCs induced a decrease in the activity of the MPO captured, compared to HaCaT cells alone. These effects are likely attributed to specific compounds released by mdMSCs and internalized by HaCaT cells to inhibit MPO activity intracellularly.

Our study has certain limitations that merit consideration. Firstly, HaCaT cells, being a continuous line of keratinocytes, may not fully represent the behaviour of primary keratinocytes. Therefore, it would be beneficial to validate our findings using primary keratinocytes to ensure the robustness of our model. Secondly, while our current model provides a simplified in vitro representation of laminitis, future advancements could involve the development of a more sophisticated model such as an OrganoPlate 3 line tissue chip, as proposed by the Mimetas company. This advanced model could incorporate additional components such as endothelial cells and a basement membrane, along with equine keratinocytes, to better mimic the complex physiological environment of the lamellar tissue.

## 5. Conclusions

Our study sheds light on the potential mechanisms underlying the recovery of HaCaT metabolic activity. While the transfer of mitochondria may play a role, our findings highlight the significant anti-inflammatory properties of mdMSCs, implicating the NETs, MPO activity, and ROS generation. These findings hint at a promising direction for future research in cell therapy for laminitis, where mdMSCs show promise in mitigating inflammation and potentially restoring keratinocyte metabolism. This work introduces a novel in vitro model that simulates laminitis in horses, aligning with the 3R principles (replacement, reduction, and refinement) of animal research.

## Figures and Tables

**Figure 1 animals-14-02681-f001:**
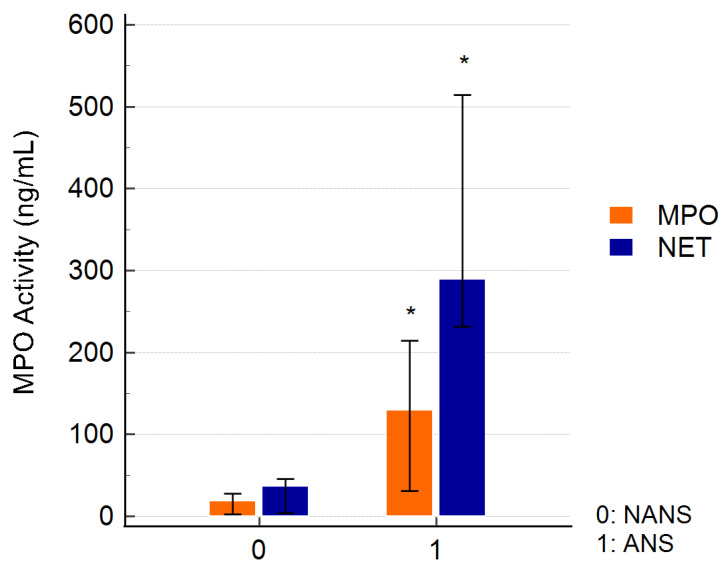
Measurement of free active MPO (MPO) and active MPO bound to the NET (NET) in the supernatants of non-activated (NANS) and activated (ANS) neutrophils with CB and fMLP. The results are presented as the medians with 95% confidence intervals of the six independent experiments. (* *p* < 0.01 vs. NANS).

**Figure 2 animals-14-02681-f002:**
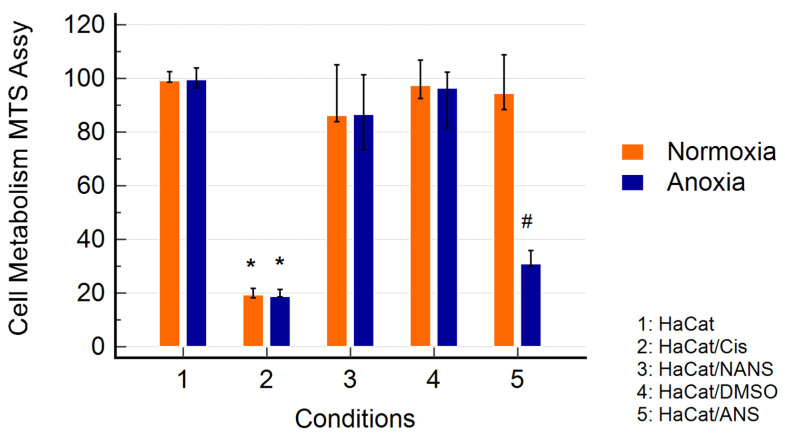
The effect of the non-activated neutrophil supernatant (NANS) and the CB/fMLP activated one (ANS) on the metabolism of HaCaTs using an MTS assay. The HaCaTs were cultured for 48 h in normoxia or anoxia (5% CO_2_) and then incubated for 24 h with a fresh medium under normoxia, followed by the measurement of metabolic activity after 4 h (MTS assay). Ctrl DMSO: a control with DMSO was used for the solubilization of CB and fMLP. The results are presented as the medians with 95% confidence intervals of the 12 experiments and are expressed as relative values (%) compared to control groups, which were standardized to 100%. (* *p* < 0.01 vs. HaCats alone; # *p* < 0.01 vs. normoxia).

**Figure 3 animals-14-02681-f003:**
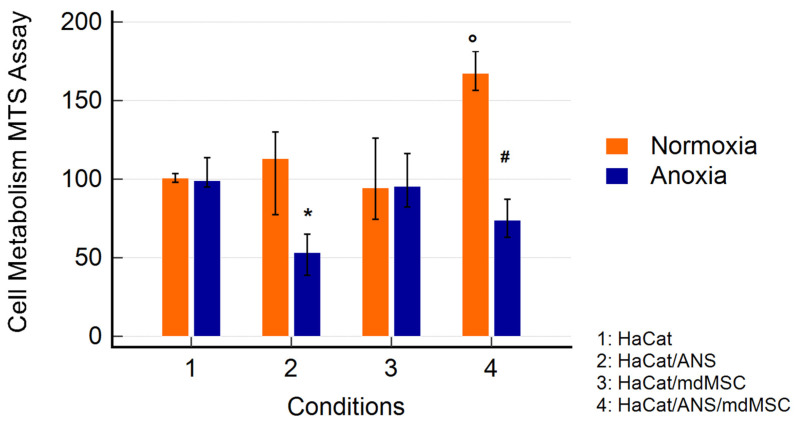
The effect of mdMSCs on HaCaTs cultured in normoxia or anoxia for 48 h in the presence of an activated neutrophil supernatant (ANS). MdMSCs were added at the reoxygenation period of 24 h, just after 48 h of normoxia or anoxia. After medium removal, the MTS solution was added for the measurement of metabolic activity. The results considered the subtraction of the metabolic response due to the mdMSCs. The results are presented as the medians with 95% confidence intervals of the 15 experiments and are expressed in the relative % vs. condition 1, set as a 100% response. (* *p* < 0.01 vs. normoxia; ° *p* < 0.01 vs. other conditions; # *p* < 0.01 between conditions 2 and 4 in anoxia).

**Figure 4 animals-14-02681-f004:**
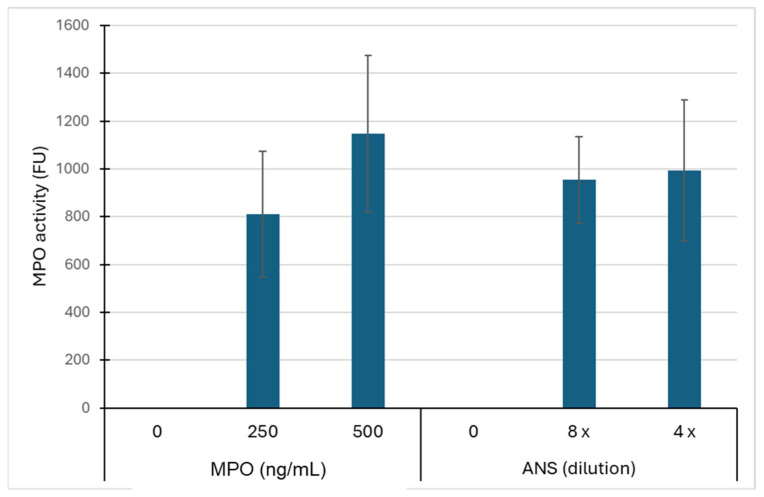
MPO activity measured for the adherent HaCaT cells (100,000/mL) after their incubation with either 250 ng/mL or 500 ng/mL of total equine MPO, or with the ANS that was diluted four or eight times into the medium. After their incubation with MPO, the cells were washed three times with a PBS buffer before the in situ measurement of the peroxidase activity. The results are expressed in fluorescence units. Mean +/− SD (*n*= 5 independent experiments).

**Figure 5 animals-14-02681-f005:**
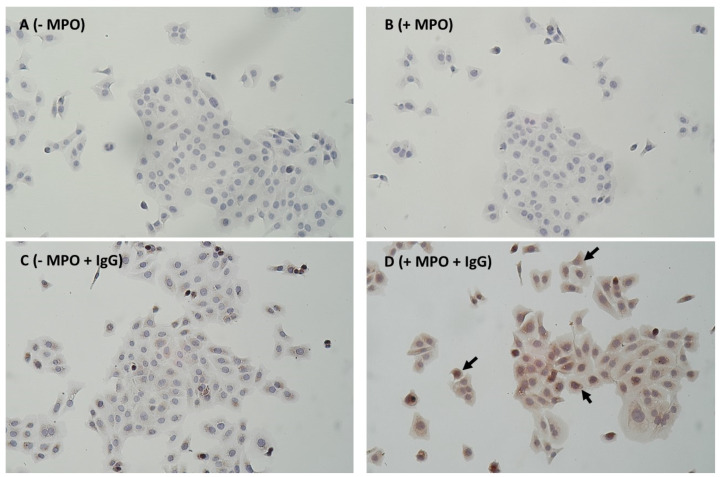
Photomicrograph of the HaCaTs pre-treated (**B**,**D**) or not (**A**,**C**) for 2 h with 500 ng/mL human MPO (+MPO). After three washings with DPBS to remove the medium with MPO, the cells were directly fixed and further stained following an immunohistochemical protocol. Panels A and B were without the primary MPO antibody, and panels C and D were with the primary MPO antibody (+IgG). After 24 h seeding, the HaCat cells formed colonies that had a squamous appearance. Positive staining to the MPO appeared in a brown colour (see arrows for example). The nuclei were stained blue by counterstaining them with hematoxylin (×100).

**Table 1 animals-14-02681-t001:** MPO activity on the adherent HaCaT cells (100,000/mL) that were preincubated with 250 ng/mL or 500 ng/mL equine MPO and then, after washing (3 times with PBS), were put in contact for 24 h with or without mdMSCs (200,000/mL). After incubation and cell washing (3 times with PBS), the peroxidase activity was measured in situ. Mean +/− SD (*n*= 6).

	In Situ Peroxidase Activity (Fluorescence Units)			
	HaCaT + MPO 250 ng/mL		HaCaT + MPO 500 ng/mL	
Horse	mdMSCs−	mdMSCs+	mdMSCs−	mdMSCs+
1	40.97	28.20	232.20	87.03
2	37.90	19.02	196.10	121.30
3	73.79	23.59	155.20	93.76
4	302.60	122.10	734.80	487.40
5	146.80	130.70	657.80	571.70
6	86.64	67.65	547.80	318.80
Mean	114.78	65.21	420.65	280.00
SD	100.13	50.53	255.93	212.94
*p*-Values		0.12		<0.01

## Data Availability

All the data generated or analysed during this study are included in this published article.

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
