# Peer review of "Revealing the Therapeutic Potential of Muscle-Derived Mesenchymal Stem/Stromal Cells: An In Vitro Model for Equine Laminitis Based on Activated Neutrophils, Anoxia–Reoxygenation, and Myeloperoxidase"

_animals, 2024, doi:10.3390/ani14182681_

Round 1
Reviewer 1 Report
Comments and Suggestions for Authors
The manuscript describes a new in vitro model simulating laminitis disease of the horse. This is a very good initiative from the authors in order to respect the 3 R principles about the biological studies using animals for scientific purposes.
Although results demonstrated significant differences as expected by the project, only 6 horses replicates are used in the study. Then, parametric statistical analysis could not be performed and non-parametric ones are. Standard deviation (SD) presented in the figures are quite high and showed a variability in the responses.Moreover, basal values are not always available as the authors indicate that control groups are taken as 100%.
Author Response
We have taken into account Reviewer 1's comments regarding the statistical analysis. The number of independent experiments has been clarified, and the results are now presented as medians with 95% confidence intervals, in line with non-parametric analysis standards.

Reviewer 2 Report
Comments and Suggestions for Authors
The work is interesting however it is noticeable that the different sections of the manuscript were written by different authors and that afterwards there was no standardization of the text, requiring a thorough revision.

/
Author Response
We have revised the references and added the complete information missing regarding the congress held in Berlin. We have also corrected all typographical errors noted.

Reviewer 3 Report
Comments and Suggestions for Authors
I've reviewed the manuscript and found some areas that require your attention to enhance its clarity and effectiveness.
Firstly, several things could be improved with the use of abbreviations. Some are used without being defined initially, and others revert to their full terms after the abbreviation is introduced. Please standardize the use of these abbreviations throughout the document.
Secondly, Figure 5, which features four panels of immunohistochemistry photomicrographs, lacks detailed descriptions necessary for non-pathologists to understand the findings. Enhancing the figure legend and manuscript text to include more comprehensive interpretation details will make the results more accessible and impactful.
Lastly, the manuscript mentions the potential use of organ-on-a-chip technology to validate the results of in vitro studies. Given that this technology has been slow to advance in veterinary medicine, could there be alternative methods currently in use for veterinary research that might be implemented more swiftly? Exploring and suggesting such alternatives could provide practical pathways for rapid validation of your findings, making the manuscript more relevant and useful.
Please review these points and make the necessary adjustments.
Author Response
As per Reviewer 3's suggestions, we have made efforts to improve the clarity of the IHC photomicrographs and have refined our use of abbreviations. Additionally, we have included a sentence discussing the potential development of organ-on-chip models in the context of our laminitis study.
